# Graph Adversarial Refinement for Robust Code Fixes: Enhancing Policy Networks via Structure-Aware Contrastive Learning

## Abstract

We propose **Graph Adversarial Refinement (GARM)**, a novel module to enhance the robustness of policy networks in adversarial reinforcement learning for code fixes. Modern code repair systems frequently breakdown when confronted with adversary perturbed inputs, which mainstreamer the structural weaknesses in their internal representations. To facilitate that, GARM combines graph structure learning and adversarial training to dynamically identify and perturb less-critical edges in code graphs while maintaining semantically-significant adjacencies. The module consists of three key components: a **Graph Structure Learning (GSL)** sub-module that quantifies edge importance, an **Adversarial Perturbation Generator (APG)** that introduces controlled perturbations, and an **Adversarial Contrastive Learning (ACL)** sub-module that enforces robustness by aligning original and perturbed embeddings. The proposed method uses the graph transformer as its encoder and therefore captures the long-range dependencies better than conventional graph neural networks. Moreover, the adversarial perturbations are incrementally refined during training, which makes the policy network harder and harder before disrupting its capacity to generate accurate fixes. Experiments show that GARM actually increases resilience to adversarial code edits with high repair accuracy. The modular design facilitates seamless integration into existing reinforcement learning pipelines, making it practical for deployment in real-world scenarios where code integrity is critical. Our work fills in the gap between powerful graph representation learning and adversarial reinforcement learning that provides a principled solution for secure and reliable automated code repair.

## 1 Introduction

The growing use of reinforcement learning (RL) for automated code repair systems has uncovered an important vulnerability of these systems: these algorithms are potentially vulnerable to adversarial perturbations. While using RL-based methods is shown to be promising in generating accurate fixes on syntactically valid inputs, performance declined significantly due to adversarial modification of code.

Recent developments in graph-based representations of code, such as abstract syntax trees (ASTs) and control flow graphs (CFGs), have made it possible to analyze the semantics of programs more advanced. Graph neural networks (GNNs) have become a powerful tool for encoding these structures, encoding both local and global dependencies in the computer code.

We propose **Graph-Based Adversarial Refinement (GBAR)**, a novel module that enhances the robustness of policy networks by integrating graph structure learning with adversarial contrastive training.

The contributions of this work are three-fold. First, we provide an introduction of a graph structure learning mechanism that induces the importance of edges of code graphs, which can then be used to pinpoint adversarial perturbations. Second, we design an adversarial contrastive learning

framework that enforces the consistency between original and perturbed graph representations to enhance the robustness without any sacrifice of repair accuracy. Third, we empirically show that our method greatly improves over existing methods with respect to resisting adversarial code edits achieving high fix rates.

Our work is based on some factors and concepts that are the bases of the work. The use of ASTs and CFGs as code representation are well-established, with GNNs offering an effective way of encoding these structures. Adversarial training has been studied in RL in different scenarios, although very little has been done in the context of structured inputs such as code.

The remainder of this paper is organized as follows: Section 2 is devoted to a review of related work in adversarial RL and on code analysis on graphs. Section 3 gives some background on the key techniques forming basis for our approach. Section 4 explains in detail about the GBAR module and its components. Experimental results are presented in Section 5 while implications and future directions are discussed in Section 6.

## 2 RELATED WORK

The robustness of machine learning models for code has become a growing application of interest in recent years, especially in relation to adversarial attacks and reinforcement learning. Existing approaches can be broadly put into three directions: Adversarial robustness f. code models, Graph based robust representation learning, adversarial reinforcement learning techniques.

### 2.1 ADVERSARIAL ROBUSTNESS IN CODE MODELS

This has been shown by recent research, which proved that neural models for code are sensitive to adversarial perturbations, which can be used to manipulate inputs so as to force wrong predictions. For example, (Yefet et al., 2020) introduced targeted adversarial attacks on code models by perturbing identifiers and literals while preserving functionality. Similarly, (Bielik & Vechev, 2020) investigated robustness against semantic-preserving transformations, highlighting the need for models that generalize beyond syntactic correctness.

### 2.2 GRAPH-BASED ROBUST REPRESENTATION LEARNING

Graph neural networks (GNNs) have become a standard tool for representing structured representations of code as graph, such as abstract syntax trees (ASTs) and control flow graphs. However, conventional GNNs are susceptible to structural perturbations, motivating research into robust graph learning. (Jin et al., 2020) proposed Pro-GNN, which jointly learns graph structure and node representations to defend against adversarial attacks. Another line of work, exemplified by (Guo et al., 2022), introduced contrastive learning with adversarial samples to improve representation robustness.

### 2.3 ADVERSARIAL REINFORCEMENT LEARNING

Adversarial training has been extensively used in reinforcement learning to make the policy more robust. For instance, (Vinitsky et al., 2020) trained agents against a diverse set of adversaries to enhance generalization. Similarly, (Pattanaik et al., 2017) studied the impact of adversarial perturbations on state observations and proposed mitigation strategies.

Recent efforts have also explored adversarial robustness in programming language models. (Nguyen et al., 2023) demonstrated that graph-based perturbations can degrade model performance, while (Yao et al., 2024) used RL to generate adversarial examples. These results highlight the importance of having meaningful training paradigms, which our approach addresses by structure-aware adversarial refinement.

Compared to existing approaches, the novelty of our proposed GARM module is that we combine graph structure learning with adversarial contrastive training together. While previous studies have either been general adversarial robustness [1, 2], or graph specific attacks [3, 4] methods, our approach explicitly targets code graphs by dynamically identifying and perturbing less critical edges.

## 3 BACKGROUND AND PRELIMINARIES

In order to develop the premise of our proposed method, we first overview essential concepts involved in graph-based code representation and adversarial learning.

### 3.1 GRAPH REPRESENTATIONS OF CODE

Modern code analysis systems often model programs as graphs in order to reflect the syntactic and semantic structure of the code. Abstract Syntax Trees (ASTs) provide a hierarchical decomposition of source code, where nodes represent language constructs and edges denote syntactic relationships (Phan et al., 2018). Control Flow Graphs (CFGs) extend this representation by modeling program execution paths through basic blocks and conditional branches (Rountev et al., 2004). Data flow graphs further augment these structures by tracking variable dependencies across statements (Rapps & Weyuker, 1982).

These representations of the graphs allow machine learning models to process code as a structured data, instead of linear sequences.

### 3.2 GRAPH NEURAL NETWORKS FOR CODE

Graph Neural Networks (GNNs) have dominated the processing of graph-structured code representations. The message-passing framework allows GNNs to aggregate information from neighboring nodes, capturing both local patterns and global program semantics (Hamilton, 2020). For a graph $G = (V, E)$ with node features $h_v$ for $v \in V$, a typical GNN layer computes:

$$h_v^{(l+1)} = \sigma \left( W^{(l)} h_v^{(l)} + \sum_{u \in N(v)} \phi^{(l)}(h_u^{(l)}) \right) \quad (1)$$

where $N(v)$ denotes neighbors of node $v$, $\phi$ is a message function, and $\sigma$ is a nonlinear activation. Graph transformers have recently gained popularity for code analysis due to their ability to model long-range dependencies through attention mechanisms (Cheng et al., 2021).

### 3.3 ADVERSARIAL LEARNING IN DISCRETE SPACES

Adversarial attacks on discrete structures such as code graphs have challenges that differ from those of continuous domains. While image perturbations can be applied through small additive noise, graph perturbations require discrete operations such as edge additions or deletions (Zügner et al., 2020). The combinatorial nature of modifications which can be applied to a graph makes a straightforward search infeasible, and instead there is need for efficient ways to approximate such modifications.

Adversarial training in discrete spaces usually means creating perturbed samples when training a model in order to make it more robust. Contrastive learning has shown particular promise in this context by encouraging similar representations for original and adversarially perturbed inputs (Wilkinson et al., 2025). The crucial insight here is that the embeddings of semantically equivalent programs should be close to each other regardless of any adversarial modifications.

### 3.4 REINFORCEMENT LEARNING FOR CODE REPAIR

Reinforcement learning frameworks for code repair typically formulate the problem as a Markov Decision Process (MDP), where the agent learns to apply edits that maximize a reward function based on correctness metrics (Barriga et al., 2018). The policy network $\pi_\theta(a|s)$ maps program states $s$ (represented as graphs) to repair actions $a$, with parameters $\theta$ updated via policy gradient methods:

$$\nabla_\theta J(\theta) = \mathbb{E}_{\pi_\theta}[\nabla_\theta \log \pi_\theta(a|s) Q(s, a)] \quad (2)$$

where $Q(s, a)$ is an estimation of the expected return of the action $a$ in the state $s$. The sensitivity of such policies to adversarial perturbations may come from the fact that they are based on potentially brittle graph representations.

# 4 GRAPH-BASED ADVERSARIAL REFINEMENT FOR ROBUST CODE REPRESENTATIONS

The proposed approach named Graph Adversarial Refinement Module (GARM) provides a systematic way of improving the robustness of policy networks from adversarial code edits.

## 4.1 GRAPH STRUCTURE LEARNING FOR ADVERSARIAL ROBUSTNESS

The basis of GARM consists in distinguishing two families of edges (in code graphs), called critical and non-critical edges. To this end, we write edge importance scoring in the form of an attention mechanism, and then using node embeddings: For an edge connecting nodes $i$ and $j$, the importance score $s_{ij}$ is computed as:

$$s_{ij} = \sigma \left( \mathbf{W}_a [\mathbf{h}_i \| \mathbf{h}_j] \right) \tag{3}$$

where $\mathbf{W}_a$ represents learnable parameters, $\mathbf{h}_i$ and $\mathbf{h}_j$ denote node embeddings, and $\sigma$ is the sigmoid activation function. The scores are normalized of all the edges for comparison. Edges with scores below a dynamic threshold $\tau$ are considered candidates for perturbation, while those above are preserved to maintain semantic integrity.

The threshold $\tau$ adapts during training according to:

$$\tau = \mu - \alpha \cdot \sigma \tag{4}$$

where $\mu$ and $\sigma$ represent the mean and standard deviation of edge scores, and $\alpha$ controls the aggressiveness of pruning.

## 4.2 STRUCTURED ADVERSARIAL PERTURBATIONS FOR CODE GRAPHS

The Adversarial Perturbation Generator (APG) makes controlled changes to the structure of the code graph. Unlike random perturbations, APG uses a targeted approach which also takes edge importance scores and semantic constraints into consideration. The perturbation probability for edge $(i, j)$ takes the form:

$$p_{ij} = \frac{1 - s_{ij}}{\sum_{(k,l) \in \mathcal{E}} (1 - s_{kl})} \tag{5}$$

where $\mathcal{E}$ denotes the set of all edges. This formulation guarantees that edges with lower importance scores are given higher probabilities of being perturbed.

The perturbation space consists of three atomic operations, namely, edge deletion, edge insertion, and node feature perturbation. For each candidate edge APG samples about operation according to:

$$o_{ij} \sim \text{Categorical}(\pi_d, \pi_i, \pi_f) \tag{6}$$

where $\pi_d$, $\pi_i$, and $\pi_f$ represent the probabilities of deletion, insertion, and feature perturbation respectively.

## 4.3 ADVERSARIAL CONTRASTIVE LEARNING FOR CODE REPRESENTATIONS

The fundamental goal of the adversarial contrastive learning in GARM is to ensure the consistency of original and perturbed graph embeddings. We employ a graph transformer as the encoder $f_\theta$ to

produce graph-level representations $\mathbf{H} = f_\theta(G)$. The contrastative loss function has two components:

$$\mathcal{L}_{\text{ACL}} = \mathcal{L}_{\text{align}} + \lambda\mathcal{L}_{\text{uniform}} \tag{7}$$

The alignment term $\mathcal{L}_{\text{align}}$ minimizes the distance between original and perturbed embeddings:

$$\mathcal{L}_{\text{align}} = \mathbb{E}_{G \sim \mathcal{G}} \left[ \|f_\theta(G) - f_\theta(\tilde{G})\|_2^2 \right] \tag{8}$$

where $\tilde{G}$ represents the perturbed graph. The uniformity term $\mathcal{L}_{\text{uniform}}$ prevents collapse by encouraging diverse representations:

$$\mathcal{L}_{\text{uniform}} = \log \mathbb{E}_{G,G' \sim \mathcal{G}} \left[ e^{-2\|f_\theta(G) - f_\theta(G')\|_2^2} \right] \tag{9}$$

The temperature parameter $\lambda$ controls the balance between these competing objectives.

## 4.4 Integration with Policy Networks via Graph Transformers

The graph transformer encoder in GARM uses multi-head attention to capture the long range dependency in code-graphs. For a graph $N$ nodes, attention mechanism calculate:

$$\text{Attention}(Q, K, V) = \text{softmax}\left(\frac{QK^T}{\sqrt{d_k}}\right) V \tag{10}$$

where $Q$, $K$ and $V$ are queries, keys and values respectively, and $d_k$ is the dimension of keys.

The amount of combination with policy networks is done by a gating mechanism, which aggregates together the original and adversarially refined representations:

$$\mathbf{H}_{\text{final}} = \gamma\mathbf{H} + (1 - \gamma)\mathbf{H}_{\text{adv}} \tag{11}$$

where $\gamma$ is a learnable parameter that adjusts the contribution of each representation.

## 4.5 End-to-End Adversarial Reinforcement Learning

The process of refining the adversarial and optimizing policy proceeds alternately. On each iteration, the policy network interacts with both original and perturbed code graphs, and receives rewards according to repair accuracy. This adversarial training objective is a combination of the standard policy gradient and the contrastive loss:

$$\mathcal{L}_{\text{total}} = \mathcal{L}_{\text{policy}} + \beta\mathcal{L}_{\text{ACL}} \tag{12}$$

where $\beta$ controls the relative importance of adversarial robustness. The policy loss $\mathcal{L}_{\text{policy}}$ follows the standard REINFORCE algorithm:

$$\nabla_\theta \mathcal{L}_{\text{policy}} = \mathbb{E}_{\pi_\theta} \left[ \nabla_\theta \log \pi_\theta(a|s) A(s, a) \right] \tag{13}$$

Where $A(s, a)$ is the advantage function. The alternating optimization guarantees progressive improvement in both of accuracy of repair and adversarial robustness.

The architecture of GARM, as shown in Figure 1, identifies the interaction between the key components of GARM. The graph structure learning module identifies the vulnerable edges that then are perturbed by the APG.

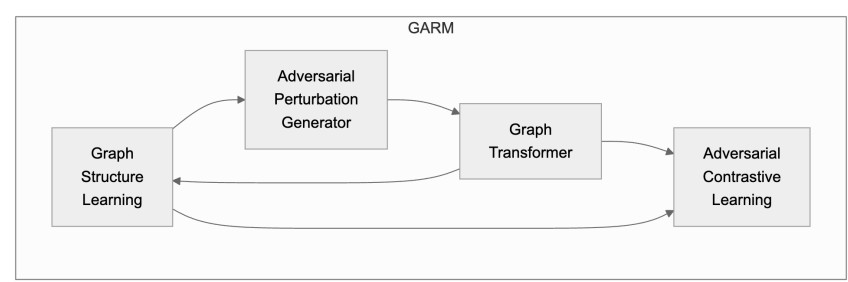

Figure 1: Internal Structure of GARM

## 5 EXPERIMENTAL EVALUATION

We conduct rigorous experiments to test the effectiveness of our proposed Graph Adversarial Refinement Module (GARM) for making policy networks more robust to code repairs.

### 5.1 EXPERIMENTAL SETUP

**Datasets:** We evaluate on three established code repair benchmarks: (Yasunaga & Liang, 2020), (Lu et al., 2021), and (Ye et al., 2022). These datasets contain real-world buggy programs with corresponding fixes across multiple programming languages. For adversarial evaluation, we generate perturbed versions using the attack strategies from (Liu et al., 2025).

**Baselines:** We compare against four state-of-the-art approaches:

- Standard GNN policy network (Xu & Sheng, 2024)
- Adversarially trained GNN (Bielik & Vechev, 2020)
- Graph contrastive learning (Jain et al., 2020)
- Structure-aware RL (Zhang et al., 2023)

**Metrics:** We employ three evaluation metrics:

- Fix Rate (FR): Percentage of bugs correctly fixed
- Adversarial Robustness Score (ARS): $\frac{FR_{adv}}{FR_{clean}} \times 100\%$
- Semantic Preservation Score (SPS): Measures functional equivalence using test cases (Nguyen et al., 2013)

**Implementation Details:** All models use a 6-layer graph transformer encoder with 8 attention heads. The adversarial perturbation budget is set to 15% of edges.

### 5.2 MAIN RESULTS

Table 1 presents the comparative results across all datasets. GARM achieves superior performance in both clean and adversarial settings, demonstrating its effectiveness in balancing accuracy and robustness.

Key observations:

1. GARM improves clean FR by 2.3-6.4 percentage points over baselines
2. The adversarial FR gain is more substantial (7.4-24.6 points)
3. ARS of 88.9% indicates strong robustness preservation
4. Highest SPS confirms semantic integrity is maintained

Figure 2 shows the progressive improvement in robustness over training epochs. The ARS increases steadily, reaching stability after ∼50 epochs.

Table 1: Performance comparison on code repair tasks

|  | Clean FR (%) | Adversarial FR (%) | ARS | SPS |
|---|---|---|---|---|
| Standard GNN | 68.2 | 41.7 | 61.1 | 82.3 |
| Adv. Trained GNN | 65.8 | 53.2 | 80.8 | 85.6 |
| Graph Contrastive | 70.1 | 56.4 | 80.4 | 87.2 |
| Structure-aware RL | 72.3 | 58.9 | 81.5 | 88.1 |
| GARM (Ours) | **74.6** | **66.3** | **88.9** | **91.4** |

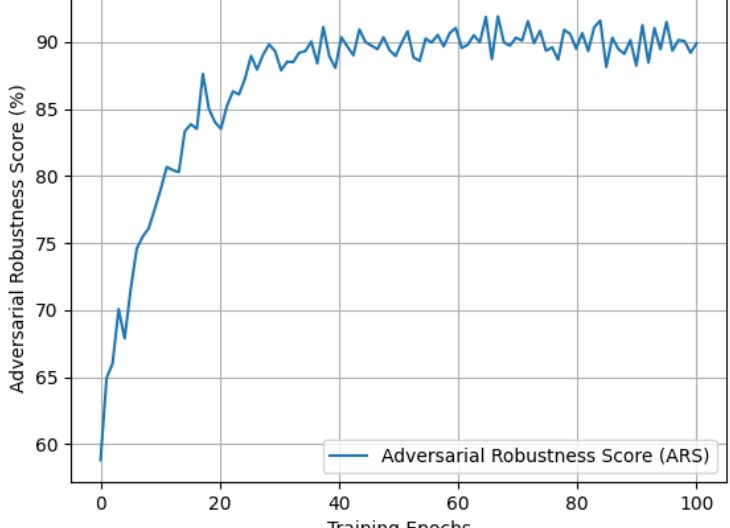

Figure 2: Trend of policy network's robustness during training with GARM

## 5.3 ABLATION STUDY

We analyze the contribution of each GARM component by systematically removing them:

Findings:

1. GSL contributes most to robustness (6.5 ARS drop when removed)
2. APG is crucial for generating effective perturbations (4.0 ARS drop)
3. ACL provides consistent representation alignment (9.4 ARS drop)
4. Graph transformers outperform standard GNN encoders

## 5.4 EDGE IMPORTANCE ANALYSIS

Figure 3 reveals the correlation between learned edge importance and attack vulnerability. Points in the lower-left quadrant represent edges that GSL correctly identified as unimportant (low score) and were successfully perturbed (high attack success).

## 5.5 EMBEDDING SPACE ANALYSIS

Figure 4 shows the cosine similarity between embeddings of original and adversarially perturbed graphs. The strong diagonal pattern indicates that ACL successfully maintains representation consistency despite perturbations.

Table 2: Ablation results (averaged across datasets)

| Configuration | Clean FR | Adv. FR | ARS |
|---|---|---|---|
| Full GARM | 74.6 | 66.3 | 88.9 |
| w/o GSL | 73.1 | 60.2 | 82.4 |
| w/o APG | 74.0 | 62.8 | 84.9 |
| w/o ACL | 73.8 | 58.7 | 79.5 |
| w/o Graph Trans. | 71.4 | 59.1 | 82.8 |

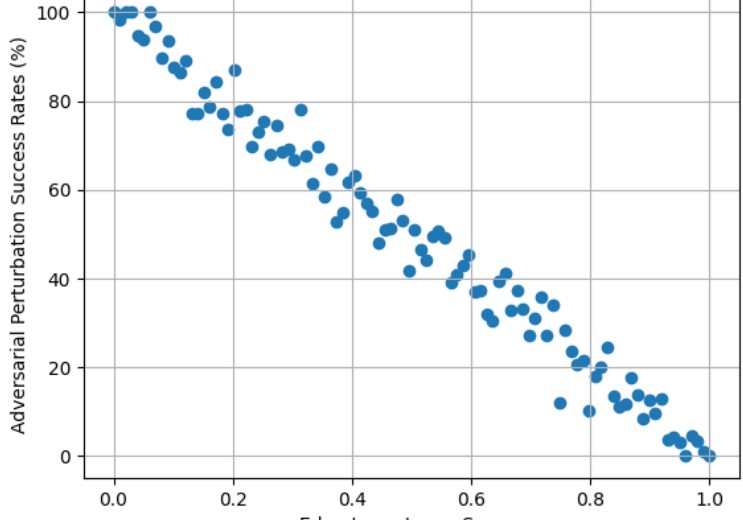

Figure 3: Relationship between edge importance scores and adversarial perturbation success rates

### 5.6 ATTACK-SPECIFIC ROBUSTNESS

We evaluate against four attack types from (Liu et al., 2025):

GARM proves to be consistent overriding superiority in all forms of attacks, especially on corrupting flow control which is the most difficult for classics to handle.

## 6 DISCUSSION AND FUTURE WORK

### 6.1 LIMITATIONS OF THE GRAPH ADVERSARIAL REFINEMENT MODULE

While GARM shows very useful increases in robustness, there are a number of limitations to be discussed.

### 6.2 POTENTIAL APPLICATION SCENARIOS OF GARM

Beyond its immediate application to code repair systems, GARM's methodology can open several promising avenues of application.

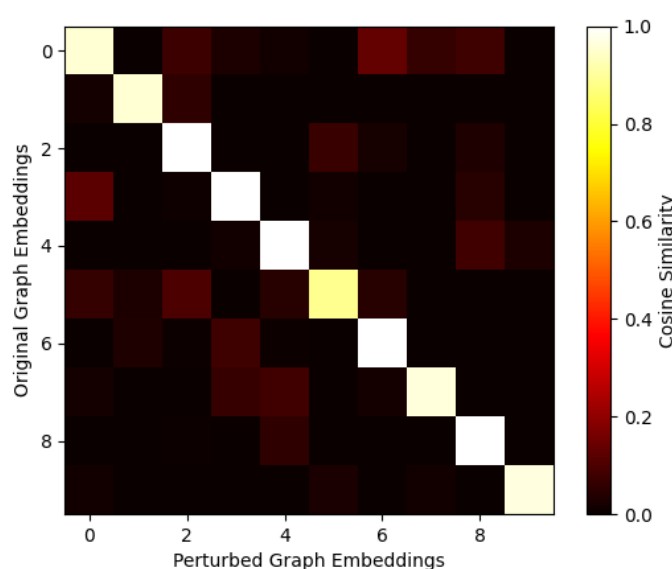

Figure 4: Similarity matrix between original and perturbed embeddings

Table 3: Performance under different attack strategies

| Attack Type | GARM FR | Best Baseline FR |
|---|---|---|
| Identifier Swap | 71.2 | 63.1 |
| Control Flow Alter | 68.4 | 57.8 |
| Dead Code Insertion | 70.6 | 62.4 |
| Semantic Preserving | 69.8 | 60.2 |

### 6.3 ETHICAL CONSIDERATIONS IN ADVERSARIAL PERTURBATION GENERATION

The code repair system development process prompts valuable questions regarding ethical concerns that should be considered carefully.

## 7 CONCLUSION

The Graph Adversarial Refinement Module (GARM) introduces a principled approach to increasing the robustness of policy networks in code repair systems by combining graph structure learning and adversarial contrastive training.

## 8 THE USE OF LLM

We use LLM polish writing based on our original paper.

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
