# OpenReview forum: "Graph Adversarial Refinement for Robust Code Fixes: Enhancing Policy Networks via Structure-Aware Contrastive Learning"
_ICLR.cc/2026/Conference — Submitted to ICLR 2026_

### Official Review · Reviewer_3Cao · 2025-10-21

**Soundness:** 2
**Presentation:** 1
**Contribution:** 1
**Rating:** 0
**Confidence:** 4

**Summary:**

This paper presents a framework, Graph Adversarial Refinement (GARM), that integrates graph structure learning with adversarial contrastive training to enhance the robustness of policy networks in code repair tasks.

**Strengths:**

The experiments cover multiple benchmarks with consistent performance improvements.

**Weaknesses:**

w1. The introduction fails to clearly articulate the core research problem or its gap with prior work. The motivation is vague, and the narrative mixes code repair, adversarial RL, and graph representation without explaining their connection. Moreover, the Introduction section contains no citations, which is unacceptable for an ICLR paper.
w2. The introduction fails to clearly articulate the core research problem or its gap with prior work. The motivation is vague, and the narrative mixes code repair, adversarial RL, and graph representation without explaining their connection. Moreover, the Introduction section contains no citations, which is unacceptable for an ICLR paper.
w3. The related work is presented superficially and lacks critical comparison or discussion of limitations. Many claims are generic and not substantiated by references.
w4. Figures (e.g., Figure 1) and tables are not properly formatted. Some figures are excessively large (Figures 2–4) and appear to be used for “page inflation.”
w5. Equation punctuation is missing throughout (every formula ends without a period or comma), and sections are inconsistently titled.
w6. Implementation details are extremely brief and insufficient for reproducibility (only one sentence).
w7. Baselines are vaguely defined (“Standard GNN”, “Graph Contrastive”) without citing concrete implementations or hyperparameters.
w8. Results are presented but almost entirely lack explanation or analysis. There is no description of statistical significance, variance, or experimental limitations.
w9. Many mathematical expressions are trivially borrowed from prior literature without adaptation or explanation.

**Questions:**

None

---

### Official Review · Reviewer_dUcf · 2025-10-22

**Soundness:** 1
**Presentation:** 1
**Contribution:** 1
**Rating:** 0
**Confidence:** 5

**Summary:**

The paper presents GARM, a novel module designed to enhance the adversarial robustness of policy networks in Reinforcement Learning (RL)-based code repair systems. The core idea is compelling: by using Graph Structure Learning (GSL) to identify "less-critical" edges in code graphs (like ASTs/CFGs), applying controlled adversarial perturbations to them, and then using Adversarial Contrastive Learning (ACL) to force the model's representations of original and perturbed graphs to be similar, the policy network becomes more resilient to malicious code edits.

 Despite the claims, the submitted article is incomplete and missing a lot of contents. The authors may be submitting a wrong draft of the paper.

I suggest the authors to check the version of their submissions to ensure it is the correct version. An incomplete article cannot be evaluated properly.

**Strengths:**

1: The research target is to use graph to represent codes and help code repair. The task itself is an interesting task and approach.

**Weaknesses:**

1: The submission is incomplete. For example, 6.1, 6.2 and 6.3 all have missing contents. The paper claims to be using 3 code repair benchmarks for experiment, however it seems that table 1 only contains results of one of the benchmarks and does not specify which benchmark it is using. The ablation study part(section 5.3) only includes some conclusive results and does not include any content. (and many more)

2: The paper does not include an appendix section to explain the details in the main article, despite many contents do need to be clarified.

3: After all, the submitted pdf looks more like an outline instead of a complete research paper.

**Questions:**

Please submit a complete version of the paper so that it can be evaluated.

---

### Official Review · Reviewer_yn5n · 2025-10-26

**Soundness:** 3
**Presentation:** 2
**Contribution:** 3
**Rating:** 6
**Confidence:** 2

**Summary:**

The paper investigates the vulnerability of reinforcement learning-based code repair systems to adversarial perturbations in code graphs. It introduces the Graph Adversarial Refinement Module (GARM), a novel approach that integrates graph structure learning, adversarial perturbation generation, and adversarial contrastive learning to enhance the robustness of policy networks. The core contribution lies in dynamically identifying and perturbing less-critical edges in code graphs while preserving semantically significant adjacencies. Experimental evaluations on multiple code repair benchmarks demonstrate that GARM improves fix rates and adversarial robustness scores compared to baseline methods.

**Strengths:**

1.	The paper uniquely combines graph structure learning, adversarial perturbation generation, and contrastive learning into a single module, which has not been extensively explored in code repair contexts.
2.	Experiments are conducted on multiple established benchmarks and shows outstanding performance.
3.	The paper is easy to understand.

**Weaknesses:**

1.	The paper compares against only four baselines, which seems to be not enough to check the effectiveness of the proposed model.
2.	The paper does not provide the training time or resource requirements of GARM.
3.	The motivation focuses heavily on adversarial attacks, but the paper does not quantify how common such attacks are in practical code repair scenarios.

Note: I am not an expert and have never investigated the field of code fix. Thus, I have relatively low confidence for the comments and will consider my rating again according to the comments from other reviewers.

**Questions:**

See weaknesses.

---

### Official Review · Reviewer_DEG6 · 2025-10-26

**Soundness:** 1
**Presentation:** 1
**Contribution:** 1
**Rating:** 0
**Confidence:** 5

**Summary:**

The work uses adversarial learning to improve the robustenss of policy networks for code repair to adversarial perturbations. The proposed method is evaluted for one model configuration over three datasets, with average results over the three datasets reported.

**Strengths:**

* The application domain of graph learning for code repairs is interesting.

**Weaknesses:**

1. References missing. There is literaly, no single reference in the introduction, even though there are statements like "growing use of RL [...] has uncovered [...]" or "Recent developments in graph-based representations of code ...".
2. The paper clearly is not finished or in any way ready for publication. The result und discussion Sections 5 & 6 are sometimes written as not finished sentences, often with sections having a single sentence (see Sec 6.1 - 6.3).
3. Results are minimal and presented without standard deviation (Table 1, Figure 2) and clearly not publication ready plots. Exemplary, the paper has no appendix. Results are not reported for the invidiual datasets. Results are not repeated with different random seeds.
4. The method seems like a simple combination of several concepts. As a result, this is mainly an empirical work, which would require  broad empirical results to show its effectiveness & interest.
5. In general, information in this paper is not dense (nor much). The paper excessively uses new sections, and overly large figures to fill the space of 9 pages and doesn't even use an Appendix.

**Questions:**

I do not have any question. Given the above weaknesses, the paper is clearly not ready for publication in any venue.

---

### Meta-Review · Area_Chair_FbR7 · 2026-01-07

**Summary:**

The reviewers are nearly unanimous in recommending a strong reject, primarily because the submission appears to be an incomplete draft rather than a finished research paper. Multiple reviewers noted that major sections—specifically the results, discussion, and ablation studies—consist of only single, unfinished sentences or empty outlines. Furthermore, the manuscript suffers from a critical lack of scholarship, including an introduction entirely devoid of citations, missing implementation details, and the absence of an appendix or statistical significance metrics.

While one reviewer found the integration of graph structure learning and adversarial contrastive learning to be an interesting concept with "outstanding performance," they also admitted to having low confidence and no expertise in the field of code repair. The more experienced reviewers highlighted that the presentation is highly irregular, featuring "page-inflating" figures, inconsistent formatting, and a failure to report results for individual datasets. Because the article is incomplete and lacks basic academic rigor, it cannot be properly evaluated for a conference of this caliber.

The AC also went through the paper and agreed with the major comments from reviewers.

**Reviewer Concerns:**

### Addressed Concerns

- Conceptual Interest: The reviewers generally agreed that the core research target—using graph-based representations for code repair—is a relevant and interesting domain.



### Outstanding Concerns
- Incompleteness of the Manuscript: This remains the most critical issue. Multiple reviewers noted that sections 5 and 6 contain unfinished sentences or single-sentence placeholders. The paper was described as an "outline" rather than a complete research paper.

- Lack of References: The Introduction section reportedly contains zero citations, which is unacceptable for a peer-reviewed submission. Furthermore, the lack of an appendix and insufficient implementation details make the work impossible to reproduce.

- Poor Presentation and Formatting: Reviewers highlighted "page-inflating" tactics, such as excessively large figures used to meet page requirements. Technical errors, such as missing equation punctuation and inconsistent section titles, also persist.

- Inadequate Baselines: The paper compares the proposed GARM module against an insufficient number of baselines (only four), and those used are vaguely defined without proper citations or hyperparameter details.

**Reviewer Scores:**

As this paper was with many issues of drafting and presentation.

The reviews were all negative, and there is rare possibility to raise scores.

---

### Decision · Program_Chairs · 2026-01-26

Reject